# Bounding the Cost of Search-Based Lifted Inference

**David Smith**
University of Texas At Dallas
800 W Campbell Rd, Richardson, TX 75080
dbs014200@utdallas.edu

**Vibhav Gogate**
University of Texas At Dallas
800 W Campbell Rd, Richardson, TX 75080
vibhav.gogate@utdallas.edu

## Abstract

Recently, there has been growing interest in systematic search-based and importance sampling-based lifted inference algorithms for statistical relational models (SRMs). These lifted algorithms achieve significant complexity reductions over their propositional counterparts by using lifting rules that leverage symmetries in the relational representation. One drawback of these algorithms is that they use an inference-blind representation of the search space, which makes it difficult to efficiently pre-compute tight upper bounds on the exact cost of inference without running the algorithm to completion. In this paper, we present a principled approach to address this problem. We introduce a lifted analogue of the propositional And/Or search space framework, which we call a lifted And/Or schematic. Given a schematic-based representation of an SRM, we show how to efficiently compute a tight upper bound on the time and space cost of exact inference from a current assignment and the remaining schematic. We show how our bounding method can be used within a lifted importance sampling algorithm, in order to perform effective Rao-Blackwellisation, and demonstrate experimentally that the Rao-Blackwellised version of the algorithm yields more accurate estimates on several real-world datasets.

## 1 Introduction

A myriad of probabilistic logic languages have been proposed in recent years [5, 12, 17]. These languages can express elaborate models with a compact specification. Unfortunately, performing efficient inference in these models remains a challenge. Researchers have attacked this problem by "lifting" propositional inference techniques; lifted algorithms identify indistinguishable random variables and treat them as a single block at inference time, which can yield significant reductions in complexity. Since the original proposal by Poole [15], a variety of lifted inference algorithms have emerged. One promising approach is the class of search-based algorithms [8, 9, 16, 19, 20, 21], which lift propositional weighted model counting [4, 18] to the first-order level by transforming the propositional search space into a smaller lifted search space.

In general, exact lifted inference remains intractable. As a result, there has been a growing interest in developing approximate algorithms that take advantage of symmetries. In this paper, we focus on a class of such algorithms, called lifted sampling methods [9, 10, 13, 14, 22] and in particular on the lifted importance sampling (LIS) algorithm [10]. LIS can be understood as a sampling analogue of an exact lifted search algorithm called probabilistic theorem proving (PTP). PTP accepts a SRM as input (as a Markov Logic Network (MLN) [17]), decides upon a lifted inference rule to apply (conditioning, decomposition, partial grounding, etc.), constructs a set of reduced MLNs, recursively calls itself on each reduced MLN in this set, and combines the returned values in an appropriate manner. A drawback of PTP is that the MLN representation of the search space is *inference unaware*; at any step in PTP, the cost of inference over the remaining model is unknown. This is problematic because unlike (propositional) importance sampling algorithms for graphical models, which can be Rao-Blackwellised [3] in a principled manner by sampling variables until the treewidth of the remaining model is bounded by a small constant (called $w$-cutset sampling [1]), it is currently not possible to Rao-Blackwellise LIS in a principled manner. To address these limitations, we make the following contributions:

1. We propose an alternate, *inference-aware* representation of the lifted search space that allows efficient computation of the cost of inference at any step of the PTP algorithm. Our approach is based on the And/Or search space perspective [6]. Propositional And/Or search associates a compact representation of a search space with a graphical model (called a pseudotree), and then uses this representation to guide a weighted model counting algorithm over the full search space. We extend this notion to Lifted And/Or search spaces. We associate with each SRM a schematic, which describes the associated lifted search space in terms of lifted Or nodes, which represent branching on counting assignments [8] to groups of indistinguishable variables, and lifted And nodes, which represent decompositions over independent and (possibly) identical subproblems. Our formal specification of lifted And/Or search spaces offers an intermediate representation of SRMs that bridges the gap between high-level probabilistic logics such as Markov Logic [17] and the search space representation that must be explored at inference time.
2. We use the intermediate specification to characterize the size of the search space associated with an SRM without actually exploring it, providing tight upper bounds on the complexity of PTP. This allows us, in principle, to develop advanced approximate lifted inference algorithms that take advantage of exact lifted inference whenever they encounter tractable subproblems.
3. We demonstrate the utility of our lifted And/Or schematic and tight upper bounds by developing a Rao-Blackwellised lifted importance sampling algorithm, enabling the user to systematically explore the accuracy versus complexity trade-off. We demonstrate experimentally that it vastly improves the accuracy of estimation on several real-world datasets.

## 2 Background and Terminology

**And/Or Search Spaces.** The And/Or search space model is a general perspective for searching over graphical models, including both probabilistic networks and constraint networks [6]. And/Or search spaces allow for many familiar graph notions to be used to characterize algorithmic complexity. Given a graphical model, $M = \langle G, \Phi \rangle$, where $G = \langle V, E \rangle$ is a graph and $\Phi$ is a set of features or potentials, and a rooted tree $T$ that spans $G$ in such a manner that the edges of $G$ that are not in $T$ are all back-edges (i.e., $T$ is a pseudo tree [6]), the corresponding *And/Or Search Space*, denoted $S_T(R)$, contains alternating levels of And nodes and Or nodes. Or nodes are labeled with $X_i$, where $X_i \in vars(\Phi)$. And nodes are labeled with $x_i$ and correspond to assignments to $X_i$. The root of the And/Or search tree is an Or node corresponding to the root of $T$.

Intuitively, the pseudo tree can be viewed as a *schematic* for the structure of an And/Or search space associated with a graphical model, which denotes (1) the conditioning order on the set $vars(\Phi)$, and (2) the locations along this ordering at which the model decomposes into independent subproblems. Given a pseudotree, we can generate the corresponding And/Or search tree via a straightforward algorithm [6] that adds conditioning branches to the pseudo tree representation during a DFS walk over the structure. Adding a cache that stores the value of each subproblem (keyed by an assignment to its context) allows each subproblem to be computed just once, and converts the search tree into a search graph. Thus the cost of inference is encoded in the pseudo tree. In Section 3, we define a lifted analogue to the backbone pseudo tree, called a *lifted And/Or schematic*, and in Section 3, we use the definition to prove cost of inference bounds for probabilistic logic models.

**First Order Logic.** An *entity* (or a constant) is an object in the model about which we would like to reason. Each entity has an associated type, $\tau$. The set of all unique types forms the set of base types for the model. A *domain* is a set of entities of the same type $\tau$; we assume that each domain is finite and is disjoint from every other domain in the model. A *variable*, denoted by a lower-case letter, is a symbolic placeholder that specifies where a substitution may take place. Each variable is associated with a type $\tau$; a valid substitution requires that a variable be replaced by an object (either an entity or another variable) with the same type. We denote the domain associated with a variable $v$ by $\Delta_v$.

We define a *predicate*, denoted by $R(t_1 :: \tau_1, \ldots, t_k :: \tau_k)$, to be a k-ary functor that maps typed entities to binary-valued random variables (also called *parameterized random variable* [15]). A *substitution* is an expression of the form $\{t_1 = x_1, \ldots, t_k = x_k\}$ where $t_i$ are variables of type $\tau_i$ and $x_i$ are either entities or variables of type $\tau_i$. Given a predicate $R$ and a substitution $\theta = \{t_1 = x_1, \ldots, t_k = x_k\}$, the application of $\theta$ to $R$ yields another k-ary functor functor with each $t_i$ replaced by $x_i$, called an *atom*. If all the $x_i$ are entities, the application yields a random variable. In this case, we refer to $\theta$ as a *grounding* of $R$, and $R\theta$ as a *ground atom*. We adopt the notation $\theta_i$ to refer to the $i$-th assignment of $\theta$, i.e. $\theta_i = x_i$.

**Statistical Relational Models** combine first-order logic and probabilistic graphical models. A popular SRM is Markov logic networks (MLNs) [17]. An MLN is a set of weighted first-order logic clauses. Given entities, the MLN defines a Markov network over all the ground atoms in its Herbrand base (cf. [7]), with a feature corresponding to each ground clause in the Herbrand base. (We assume Herbrand interpretations throughout this paper.) The weight of each feature is the weight of the corresponding first-order clause. The probability distribution associated with the Markov network is given by: $P(\mathbf{x}) = \frac{1}{Z}\exp(\sum_i w_i n_i(\mathbf{x}))$ where $w_i$ is the weight of the $i$th clause and $n_i(\mathbf{x})$ is its number of true groundings in $\mathbf{x}$, and $Z = \sum_{\mathbf{x}}\exp(\sum_i w_i n_i(\mathbf{x}))$ is the partition function. In this paper, we focus on computing $Z$. It is known that many inference problems over MLNs can be reduced to computing $Z$.

**Probabilistic Theorem Proving (PTP)** [9] is an algorithm for computing $Z$ in MLNs. It lifts the two main steps in propositional inference: conditioning (Or nodes) and decomposition (And nodes). In lifted conditioning, the set of truth assignments to ground atoms of a predicate $R$ are partitioned into multiple parts such that in each part (1) all truth assignment have the same number of true atoms and (2) the MLNs obtained by applying the truth assignments are identical. Thus, if $R$ has $n$ ground atoms, the lifted search procedure will search over $O(n + 1)$ new MLNs while the propositional search procedure will search over $O(2^n)$ MLNs, an exponential reduction in complexity. In lifted decomposition, the MLN is partitioned into a set of MLNs that are not only identical (up to a renaming) but also disjoint in the sense that they do not share any ground atoms. Thus, unlike the propositional procedure which creates $n$ disjoint MLNs and searches over each, the lifted procedure searches over just one of the $n$ MLNs (since they are identical). Unfortunately, lifted decomposition and lifted conditioning cannot always be applied and in such cases PTP resorts to propositional conditioning and decomposition. A drawback of PTP is that unlike propositional And/Or search which has tight complexity guarantees (e.g., exponential in the treewidth and pseudotree height), there are no (tight) formal guarantees on the complexity of PTP.[1] We address this limitation in the next two sections.

## 3 Lifted And/Or Schematics

Our goal in this section is to define a lifted analogue the *pseudotree* notion employed by the propositional And/Or framework. The structure must encode (1) all information contained in a propositional pseudotree (a conditioning order, conditional independence assumptions), as well as (2) additional information needed by the PTP algorithm in order to exploit the symmetries of the lifted model. Since the symmetries that can be exploited highly depend on the amount of evidence, we encode the

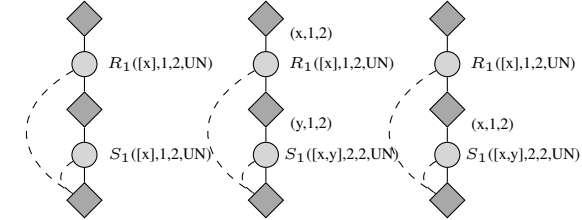

Figure 1: Possible schematics for (a) $R(x) \vee S(x)$, (b) $R(x) \vee S(x,y)$ and (c) $R(x) \vee R(y) \vee S(x,y)$, $\Delta_x = \Delta_y = 2$. $UN$ stands for unknown. Circles and diamonds represent lifted Or and And nodes respectively.

SRM after evidence is instantiated, via a process called shattering [2]. Thus, while a pseudotree encodes a graphical model, a schematic encodes an (SRM, evidence set) pair.

**Definition** A *lifted Or node* is a vertex labeled by a $6 - tuple\ \langle R, \Theta, \alpha, i, c, t \rangle$, where (1) $R$ is a $k$-ary predicate, (2) $\Theta$ is a set of valid substitutions for $R$, (3) $\alpha \in \{1, \ldots, k\}$, represents the *counting argument* for the predicate $R(t_1 :: \tau_1, \ldots, t_k :: \tau_k)$ and specifies a domain $\tau_\alpha$ to be counted over, (4) $i$ is an identifier of the block of the partition being counted over, (5) $c \in \mathbb{Z}^+$ is the number of entities in block $i$, and (6) $t \in \{True, False, Unknown\}$ is the truth value of the set of entities in block $i$.

**Definition** A *lifted And node* is a vertex labeled by $F$, a (possibly empty) set of formulas, where a *formula* $f$ is a pair $(\{(O, \theta, b)\}, w)$, in which $O$ is a lifted Or node $\langle R, \Theta, \alpha, i, c, t \rangle$, $\theta \in \Theta$, $b \in \{True, False\}$, and $w \in \mathbb{R}$. Formulas are assumed to be in clausal form.

**Definition** A *lifted And/Or schematic*, $S = \langle V_S, E_S, v_r \rangle$, is a rooted tree comprised of lifted Or nodes and lifted And nodes. $S$ must obey the following properties:

- Every lifted Or node $O \in V_S$ has a single child node $N \in V_S$.
- Every lifted And node $A \in V_S$ has a (possibly empty) set of children $\{N_1, \ldots, N_n\} \subset V_S$.

- For each pair of lifted Or nodes $O, O' \in V_S$, with respective labels $\langle R, \Theta, \alpha, i, c, t \rangle$, $\langle R', \Theta', \alpha', i', c', t' \rangle$, $(R, i) \neq (R', i')$. Pairs $(R, i)$ uniquely identify lifted Or nodes.
- For every lifted Or node $O \in V_S$ with label $\langle R, \Theta, \alpha, i, c, t \rangle$, $\forall \theta \in \Theta$, $\forall \alpha' \neq \alpha$, either (1) $\Delta_{\theta_{\alpha'}} = 1$, or (2) $\theta_{\alpha'} \in X$, where $X$ has appeared as the decomposer label [9] of some edge in $path_S(O, v_r)$.
- For each formula $f_i = (\{(O, \theta, b)\}, w)$ appearing at a lifted And node $A$, $\forall O \in \{(O, \theta, b)\}$, $O \in path_S(v_r, A)$. We call the set of edges $\{(O, A) \mid O \in \text{Formulas}(A)\}$ the *back edges* of $S$.
- Each edge between a lifted Or node $O$ and its child node $N$ is unlabeled. Each edge between a lifted And node $A$ and its child node $N$ may be (1) unlabeled or (2) labeled with a pair $(X, c)$, where $X$ is a set of variables, called a decomposer set, and $c \in \mathbb{Z}^+$ is the the number of equivalent entities in the block of $x$ represented by the subtree below. If it is labeled with a decomposer set $X$ then (a) for every substitution set $\Theta$ labeling a lifted Or node $O'$ appearing in the subtree rooted at $N$, $\exists i \ s.t \ . \forall \theta \in \Theta, \theta_i \in X$ and (b) $\forall$ decomposer sets $Y$ labeling edges in the subtree rooted at $N$, $Y \cap X = \varnothing$.

The lifted And/Or Schematic is a general structure for specifying the inference procedure in SRMs. It can encode models specified in many formats, such as Markov Logic [17] and PRV models [15]. Given a model and evidence set, constructing a schematic conversion into a canonical form is achieved via shattering [2, 11], whereby exchangeable variables are grouped together. Inference only requires information on the size of these groups, so the representation omits information on the specific variables in a given group. Figure 1 shows And/Or schematics for three MLNs.

---

**Algorithm 1** Function evalNode(And)

1: **Input:** a schematic, $T$ with And root node, a counting store $cs$
2: **Output:** a real number, $w$
3: $N = \text{root}(T)$
4: **for** formula $f \in N$ **do**
5: $\quad w = w \times \text{calculateWeight}(f, cs)$
6: **for** child $N'$ of $T$ **do**
7: $\quad cs' = \text{sumOutDoneAtoms}(cs, N)$
8: $\quad$ **if** $(N, N')$ has label $\langle V, b, c_b \rangle$ **then**
9: $\quad\quad$ **if** $\nexists \langle (V, b), cc \rangle \in cs \ s.t. \ v \in V$ **then**
10: $\quad\quad\quad cs'' = cs' \cup \langle (V, b), \langle \{\}, \{(\{\}, c_b)\} \rangle \rangle$
11: $\quad\quad \langle P, M \rangle = \text{getCC}(V, b, cs'')$ //get cc for V
12: $\quad\quad$ **for** assignment $(a_i, k_i) \in M$ **do**
13: $\quad\quad\quad$ //give v its own entry in cs
14: $\quad\quad\quad cs''' = \text{updateCCAtDecomposer}(cs'', V, v, (a_i, 1))$
15: $\quad\quad\quad w = w \times \text{evalNode}(N', cs''')^{k_i}$
16: $\quad$ **else**
17: $\quad\quad w = w \times \text{evalNode}(N', cs)$
18: **return** $w$

**Algorithm 2** Function evalNode(Or)

1: **Input:** a schematic, $T$ with Or Node root, a counting store $cs$
2: **Output:** a real number, $w$
3: **if** $(\langle \text{root}(T), cs \rangle, w) \in$ cache **then return** $w$
4: $\langle R, \Theta, \alpha, b, c, t, P \rangle = \text{root}(T)$
5: $T' = \text{child}(\langle R, \Theta, \alpha, b, c, t, \rangle, T)$
6: $V = \{v \mid \theta \in \Theta, \theta_\alpha = v\}$
7: $\langle P, \{\langle a_i, k_i \rangle\} \rangle = \text{getCC}(\mathcal{V}, b)$
8: $w = 0$
9: **if** $t \in \{True, False\}$ **then**
10: $\quad cs' = \text{updateCC}(\langle P, M \rangle, R, t_v)$
11: $\quad w = \text{evalNode}(T', cs')$
12: **else**
13: $\quad \text{assigns} = \{\{v_1, \ldots, v_n\} \mid v_i \in \{0, \ldots, k_i\}\}$
14: $\quad$ **for** $\{v_1, \ldots, v_n\} \in assigns$ **do**
15: $\quad\quad cs' = \text{updateCC}(\langle P, M \rangle, R, \{v_1, \ldots, v_n\})$
16: $\quad\quad w = w + \left( \left( \prod_{i=1}^n \binom{k_i}{v_i} \right) \text{evalNode}(T', cs') \right)$
17: insertCache$(\langle R, \Theta, \alpha, b, c, t, P \rangle, w)$
18: **return** $w$

---

**3.1 Lifted Node Evaluation Functions-**We describe the inference procedure in Algorithms 1 and 2. We require the notion of a counting store in order to track counting assignments over the variables in the model. A counting store is a set of pairs $\langle (V, i), cc \rangle$, where $V$ is a set of variables that are counted over together, $i$ is a block identifier, and $cc$ is a counting context. A counting context (introduced in [16]), is a pair $\langle Pr, M \rangle$, where $Pr$ is a list of $m$ predicates and $M : \{True, False\}^m \to k$, is a map from truth assignments to $Pr$ to a non-negative integer denoting the count of the number of entities in the $i$-th block of the partition of each $v \in V$ that take that assignment. We initialize the algorithm by a call to Algorithm 1 with an appropriate schematic $S$ and empty counting store.

The lifted And node function (Algorithm 1) first computes the weight of any completely conditioned formulas. It then makes a set of evalNode calls for each of its children $O$; if $(A, O)$ has decomposer label $V$, it makes a call for each assignment in each block of the partition of $V$; otherwise it makes a single call to $O$. The algorithm takes the product of the resulting terms along with the product of the weights and returns the result. The lifted Or node function (Algorithm 2) retrieves the set of all assignments previously made to its counting argument variable set; it then makes an evalNode call to its child for each completion to its assignment set that is consistent with its labeled truth value, and takes their weighted sum, where the weight is the number of symmetric assignments represented by each assignment completion.

The overall complexity of depends on the number of entries in the counting store at each step of inference. Note that Algorithm 1 reduces the size of the store by summing out over atoms that leave context. Algorithm 2 increases the size of the store at atoms with unknown truth value by splitting the current assignment into True and False blocks w.r.t. its atom predicate. Atoms with known truth value leave the size of the store unchanged.

## 4 Complexity Analysis

Algorithms 1 and 2 describe a DFS-style traversal of the lifted search space associated with $S$. As our notion of complexity, we are interested in specifying the maximum number of times any node $V_S \in S$ is replicated during instantiation of the search space. We describe this quantity as $SS_N(S)$. Our goal in this section is to define the function $SS_N(S)$, which we refer to as the *induced lifted width* of $S$.

**4.1 Computing the Induced Lifted Width of a Schematic-**In the propositional And/Or framework, the inference cost of a pseudotree $T$ is determined by $D_R$, the tree decomposition of the graph $G = \langle Nodes(T), BackEdges(T) \rangle$ induced by the variable ordering attained by traversing $T$ along any DFS ordering from root to leaves. [6]. Inference is $O(exp(w))$, where $w$ is the size of the largest cluster in $D_R$. The analogous procedure in lifted And/Or requires additional information be stored at each cluster. Lifted tree decompositions are identical to their propositional counterparts with two exceptions. First, each cluster $C_i$ requires the ordering of its nodes induced by the original order of $S$. Second, each cluster $C_i$ that contains a node which occurs after a decomposer label requires the inclusion of the decomposer label. Formally:

**Definition** The *tree sequence* $T_S$ associated with schematic $S$ is a partially ordered set such that: (1) $O \in Nodes(S) \Rightarrow O \in T_S$, (2) $(A, N)$ with label $l \in Edges(S) \Rightarrow (A, l) \in T_S$, and (3) $Anc(N_1, N_2, S) \Rightarrow N_1 < N_2 \in T_S$.

**Definition** The *path sequence* $P$ associated with tree sequence $T_S$ of schematic $S$ is any totally ordered subsequence of $T_S$.

**Definition** Given a schematic $S$ and its tree sequence $T_S$, the *Lifted Tree Decomposition* of $T_S$, denoted $D_S$, is a pair $(\mathcal{C}, T)$ in which $\mathcal{C}$ is a set of path sequences and $T$ is a tree whose nodes are the members of $\mathcal{C}$ satisfying the following properties: (1) $\forall (O, A) \in BackEdges(P), \exists i$ s.t. $O, A \in C_i$, (2) $\forall i, j, k$ s.t $C_k \in Path_T(C_i, C_j), C_i \cap C_j \subseteq C_k$, (3) $\forall A \in T_S, O \in C_i, A < O \Rightarrow A \in C_i$.

Given the partial ordering of nodes defined by $S$, each schematic $S$ induces a unique Lifted Tree Decomposition, $D_S$. Computing $SS_N(S)$ requires computing $\max_{C_i \in \mathcal{C}} SS_C(C_i)$. There exists a total ordering over the nodes in each $C_i$; hence the lifted structure in each $C_i$ constitutes a path. We take the lifted search space generated by each cluster $C$ to be a tree; hence computing the maximum node replication is equivalent to computing the number of leaves in $SS_C$.

In order to calculate the induced lifted width of a given path, we must first determine which Or nodes are counted over dependently. Let $V_C = \{v \mid \langle R, \Theta, \alpha, i, c, t \rangle \in C, \theta \in \Theta, \theta_\alpha = v\}$ be the set of variables that are counted over by an Or node in cluster $C$. Let $\mathcal{V}_C$ be a partition of $V_C$ into its dependent variable counting sets; i.e. define the binary relation $C_S = \{(v_1, v_2) \mid \exists \langle R, \Theta, \alpha, i, c, t \rangle \in V_S \; s.t \; \exists \theta, \theta' \in \Theta, \theta_\alpha = v_1, \theta'_\alpha = v_2\}$. Then $V = \{v' \mid (v, v') \in C_S^+\}$, where $C_S^+$ is the transitive closure of $C_S$. Let $\mathcal{V}_C = \{V_j \mid v_1, v_2 \in V_j \iff (v_1, v_2) \in C_S^+\}$. Variables that appear in a set $V_j \in \mathcal{V}_C$ refer to the same set of objects; thus all have the same type $\tau_j$ and they all share the same partition of the entities of $T_j$. Let $\mathcal{P}_j$ denote the partition of the entities of $T_j$ w.r.t variable set $\mathcal{V}_j$. Then each block $p_{ij} \in \mathcal{P}_j$ is counted over independently (we refer to each $p_{ij}$ as a *dependent counting path* ). Thus we can calculate the total leaves corresponding to cluster $C$ by taking the product of the leaves of each $p_{ij}$ block:

$$SS_C(C) = \prod_{V_j \in \mathcal{V}_C} \prod_{p_{ij} \in P_j} SS_p(p_{ij}) \tag{1}$$

Analysis of lifted Or nodes that count over the same block $p_{ij}$ depends on the structure of the decomposers sets over the structure. First, we consider the case in which $C$ contains no decomposers.

**4.2 Lifted Or Nodes with No Decomposer-**Consider $OR_{C,V_j,i}$, the sequence of nodes in $C$ that perform conditioning over the $i$-th block of the partition of the variables in $V_j$. The nodes in $OR_{C,V_j,i}$ count over the same set of entities. A conditioning assignment at $O$ assigns $c_t \in \{0 \ldots c\}$ entities to $True$ and $c_f = c - c_t$ entities to $False$ w.r.t. its predicate, breaking the symmetry over the $c$ elements in the block. Each $O' \in OR_{p,V_j,i}$ that occurs after $O$ must perform counting over two sets of size $c_t$ and $c_f$ separately. The number of assignments for block $\{V_j, i\}$ grows exponentially with the number of ancestors counting over $\{V_j, i\}$ whose truth value is unknown. Formally, let $c_{ij}$ be the size of the $i$-th block of the partition of $\mathcal{V}_j$, and let $n_{ij} = |\{O \mid O \in OR_{C,V_j,i}, N = \langle R, \Theta, \alpha, i, c, unknown \rangle\}|$. For an initial domain size $c_{ij}$ and predicate count $n_{ij}$, we must compute the number of possible ways to represent $c_{ij}$ as a sum of $2^{n_{ij}}$ non-negative integers. Define $k_{ij} = 2^{n_{ij}}$. We can count the number of leaf nodes generated by counting the number of weak compositions of $c_{ij}$ into $k_{ij}$ parts. Thus the number of search space leaves corresponding to $p_{ij}$ generated by $C$ is:

$$SS_p(p_{ij}) = W(c_{ij}, k_{ij}) = \binom{c_{ij} + k_{ij} - 1}{k_{ij} - 1} \tag{2}$$

**Example** Consider the example in Figure 1(a). There is a single path from the root to a leaf. The set of variables appearing on the path, $\mathcal{V} = \{x\}$, and hence the partition of $\mathcal{V}$ into variables that are counted over together yields $\{\{x\}\}$. Thus $n_{1,1} = |\{(R_1(2, Un), S_1(2, Un))\}| = 2$, $c_{1,1} = 2$, and $k_{1,1} = 4$. So we can count the leaves of the model by the expression $\binom{2+4-1}{4-1} = \frac{5!}{3!2!} = 10$.

**4.3 Lifted Or Nodes with Decomposers-** To determine the size of the search tree induced by a subsequence $P$ that contains decomposers, we must consider whether the counting argument of each Or node is decomposed on.

**4.3.1 Lifted Or Nodes with Decomposers as Non Counting Arguments**

We first consider the case when $OR_{C,V_j,i}$ contains decomposer variables as non-counting arguments. For each parent-to-child edge (A,N,label l), Algorithm 1 generates a child for each non-zero assignment in the counting store containing the decomposer variable. If a path subsequence over variable $v$ of initial domain $c$ has $n$ Or nodes, $k$ of which occur below the decomposer label, then we can compute the number of assignments in the counting store at each decomposer as $2^{n-k}$. Further, we can compute the number of non-zero leaves generated by each assignment can be computed as the difference in leaves from the model over $n$ Or nodes and the model over $k$ Or nodes. Hence the resulting model has $\left(2^{n-k}\right)\left(\binom{c+2^n-1}{2^n-1} - \binom{c+2^k-1}{2^k-1}\right)$ leaves. This procedure can be repeated by recursively applying the rule to split each weak composition into a difference of weak compositions for each decomposer label present in the subsequence under consideration (Algorithm 3). The final result is a polynomial in $c$, which, when given a domain size, returns the number of leaves generated by the path subsequence.

---

**Algorithm 3** Function countPathLeaves

```
1: Input: a subsequence path P
2: Output: f(x) : ℤ⁺ → ℤ⁺, where x is a domain size and f(x)
   is the number of search space leaves generated by P
3: //we represent the recursive polynomial
   a(wc₁ - wc₂) as a triple (a,wc₁,wc₂),
   where a ∈     ℤ, and wc₁,wc₂ are either weak
   compositions (base case) or triples of this
   type (recursive case)
4: type WCP = WC INT | WCD (INT,WCP,WCP)
5: //evalPoly constructs the polynomial
6: function MAKEPOLY((WC n), (t, a, s))
7:     return WCD (n/(2^(t-a)), WC n, WC (n − 2^(t-a)))
8: function MAKEPOLY((WCD (c, wc₁, wc₂)), (t, a, s))
9:     return WCD(a, makePoly wc₁ (t, a, s), makePoly wc₂ (t−
       s, a − s, s))
10: //applyDec divides out the Or nodes with
    counting variables that are decomposers
11: function APPLYDEC(d,(WC a))
12:     return WC (a/(2^d))
13: function APPLYDEC(d,(WCD (a,b,c)))
14:     return WCD (a,applyDec d b,applyDec d c)
15: //evalPoly creates a function that takes a
    domain and computes the differences of the
    constituent weak compositions
16: function EVALPOLY((WCD (a,b,c)),x)
17:     return a × (evalPoly b x - evalPoly c x)
18: function EVALPOLY((WC a),x)
19:     return (x+a-1 choose a-1)
20: t = totalOrNodes(P)
21: dv = orNodesWithDecomposerCountingArgument(P)
22: poly = WC 2^t; orNodesAbove=0;orNodesBetween=0
23: for N of P do
24:     if N = (A, ⟨v, p, c⟩) then
25:         poly = makePoly poly (t,orNodesAbove,orNodesBetween)
26:         orNodesBetween=0
27:     else
28:         orNodesAbove++;orNodesBetween++
    return 2^(dv) × evalPoly (applyDec dv poly)
```

---

**Example** Consider the example in Figure 1(c). Again there is a single path from the root to a leaf. The set of variables appearing on the path is $V = \{x, y\}$. The partition of $V$ into variables that are counted over together yields $\mathcal{V} = \{\{x, y\}\}$. Algorithm 3 returns the polynomial $f(x) = 2(W(x, 4) - W(x, 2))$. So the search space contains $2(\binom{2+4-1}{4-1} - \binom{2+2-1}{2-1}) = 14$ leaves.

**4.3.2 Lifted Or Nodes with Decomposers as Counting Arguments**

The procedure is similar for the case when $P$ contains Or nodes that count over variables that have been decomposed one addition. Or nodes that count over a variable that has previously appeared as the decomposer label of an ancestor in the path have a domain size of 1 and hence always spawn $W(1, 2) = 2$ children instead of $W(x, 2)$ children. If there are $d$ Or nodes in $P$ that count over decomposed variables, we must divide the $k$ term of each weak composition in our polynomial by $2^d$. Lines $11 - 14$ of Algorithm 3 perform this operation.

**Example** Consider the example shown in Figure 1(b). Again there is one path from the root to leaf, with $V = \{x, y\}$; partitioning $V$ into sets of variables that are counted over together yields $\mathcal{V} = \{\{x\}, \{y\}\}$. Thus $n_{1,1} = |\{(R_1(2, Un))\}| = 1$, $c_{1,1} = 2$, and $k_{1,1} = 2$. Similarly, $n_{2,1} = |\{S_1(2, Un)|]| = 1$, $c_{2,1} = 2$, and $k_{2,1} = 2$. Algorithm 3 returns the constant functions $f_1(x) = f_2(x) = 2 \times W(x, 1) = 2$. Equation 1 indicates that we take the product of these functions. So the search space contains 4 leaves regardless of the domain sizes of $x$ and $y$.

**4.4 Overall Complexity-** Detailed analysis, as well as a proof of correctness of Algorithm 3 is given in the supplemental material section. Here we give general complexity results.

**Theorem 4.1** *Given a lifted And/Or Schematic $S$ with associated Tree Decomposition $D_S = (\mathcal{C}, T)$, the overall time and space complexity of inference in $S$ is $O(max_{C_i \in \mathcal{C}} SS_C(C_i))$.*

## 5 An Application: Rao-Blackwellised Importance Sampling

Rao-Blackwellisation [1, 3] is a variance-reduction technique which combines exact inference with sampling. The idea is to partition the ground atoms into two sets: a set of atoms, say $\mathbf{X}$ that will be sampled and a set of atoms that will be summed out analytically using exact inference techniques, say $\mathbf{Y}$. Typically, the accuracy (variance decreases) improves as the cardinality of $\mathbf{Y}$ is increased. However, so does the cost of exact inference, which in turn decreases the accuracy because fewer samples are generated. Thus, there is a trade-off.

Rao-Blackwellisation is particularly useful in lifted sampling schemes because subproblems over large sets of random variables are often tractable (e.g. subproblems containing $2^n$ assignments can often be summed out in $O(n)$ time via lifted conditioning, or in $O(1)$ time via lifted decomposition). The approach presented in Section 3 is ideal for this task because Algorithm 3 returns a function that is specified at the *schematic level* rather than the *search space level*. Computing the size of the remaining search space requires just the evaluation of a set of polynomials. In this section, we introduce

---

**Algorithm 4** Function makeRaoFunction

1: **Input:** a schematic $S$
2: **Output:** $f(x) : CS \rightarrow \mathbb{Z}^+$
3: find the clusters of $S$
4: $(\mathcal{C}, \mathcal{T})$ = findTreeDecomposition($S$)
5: $sizef = \{\}$
6: **for** $\mathcal{C}_i$ of $\mathcal{C}$ **do**
7: $\quad P$ = dependentCountingPaths($\mathcal{C}_i$)
8: $\quad cf = \{\}$
9: $\quad$ **for** $(V_j, \mathcal{P}_j)$ of $\mathcal{P}$ **do**
10: $\quad\quad f_j$ = countPathLeaves($\mathcal{P}_j$)
11: $\quad\quad cf$.append($\langle V_J, f_j \rangle$)
12: $\quad sizef$.append($cf$)
    **return** $sizef$

---

**Algorithm 5** Function evalRaoFunction

1: **Input:** a counting store, $cs$, a list of list of size functions, $sf$
2: **Output:** $s \in \mathbb{Z}^+$, the cost of exact inference
3: $clusterCosts = \{\}$
4: **for** $cf_i$ of $sf$ **do**
5: $\quad clusterCost = 1$
6: $\quad$ **for** $\langle V_j, f_j \rangle$ of $cf_i$ **do**
7: $\quad\quad assigns = getCC(V_j)$
8: $\quad\quad$ **for** $s_k$ of $assigns$ **do**
9: $\quad\quad\quad clusterCost = clusterCost \times f_j(s_k)$
10: $\quad clusterCosts$.append($clusterCost$)
    **return** max($clusterCosts$)

---

our sampling scheme, which adds Rao-Blackwellisation to lifted importance sampling (LIS) (as detailed in [9, 10]). Technically, LIS is a minor modification of PTP, in which instead of searching over all possible truth assignments to ground atoms via lifted conditioning, the algorithm generates a random truth assignment (lifted sampling), and weighs it appropriately to yield an unbiased estimate of the partition function.

**5.1 Computing the size bounding function-**Given a schematic $S = \langle V_S, E_S, v_r \rangle$ to sample, we introduce a preprocessing step that constructs a size evaluation function for each $v \in V_S$. Algorithm 4 details the process of creating the function for one node. It takes as input the schematic $S$ rooted at $v$. It first finds the tree decomposition of $S$. The algorithm then finds the dependent paths in each cluster; finally, it applies Algorithm 3 to each dependent path and wraps the resulting function with the variable dependency. It returns a list of list of (variable,function) pairs.

**5.2 Importance Sampling at lifted Or Nodes-**Importance sampling at lifted Or nodes is similar to its propositional analogue. Each lifted Or node is now specified by an 8-tuple $\langle R, \Theta, \alpha, i, c, t, Q, sf \rangle$, in which $Q$ is the proposal distribution for $(R, i)$, and $sf$ is the output of Algorithm 4. The sampling algorithm takes an additional input, $cb$, specifying the complexity bound for Rao-Blackwellisation. Given an or Node where $t =$unknown, we first compute the cost of exact inference.

Algorithm 5 describes the procedure. It takes as input (1) the list of lists $sf$ output by Algorithm 4, and (2) the counting store, detailing the counting assignments already made by the current sample. For each sublist in the input list, the algorithm evaluates each (variable,function) pair by (1) retrieving the list of current assignments from the counting store, (2) evaluating the function for the domain size of each assignment, and (3) computing the product of the results. Each of these values represents a bound on the cost of inference for a single cluster; Algorithm 5 returns $c$, the maximum of this list.

If $c <= cb$ we call $evalNode(S)$; otherwise we sample assignment $i$ from $Q$ with probability $q_i$, update the counting store with assignment $i$, and call $sampleNode(S')$, where $S'$ is the child schematic, yielding estimate $\hat{w}$ of the partition function of $S'$. We then return $\hat{\delta}_S = \frac{\hat{w}}{q_i}$ as the estimate of the partition function at $S$.

**5.3 Importance Sampling at lifted And Nodes-**Importance sampling at lifted And nodes differs from its propositional counterpart in that a decomposer labeled edge $(A, T)$ represents $d$ distributions

that are not only independent but also *identical*. Let $A$ be a lifted And node that we wish to sample, with children $S_1, \ldots, S_k$, with corresponding decomposer labels $d_1 \ldots d_k$ (for each edge with no decomposer label take $d_i = 1$). Then the estimator for the partition function at $A$ is: $\hat{\delta}_A = \prod_{i \in \{1..k\}} \prod_{j \in \{1..d_i\}} \delta_{T_i}$.

## 6 Experiments

We ran our Rao-Blackwellised Importance Sampler on three benchmark SRMs and datasets: (1) The friends, smokers and Asthma MLN and dataset described in [19], (2) The webKB MLN for collective classification and (3) The Protein MLN, in which the task is to infer protein interactions from biological data. All models are available from `www.alchemy.cs.washington.edu`.

**Setup.** For each model, we set 10% randomly selected ground atoms as evidence, and designated them to have $True$ value. We then estimated the partition function via our Rao-Blackwellised sampler with complexity bounds $\{0, 10, 100, 1000\}$ (bound of 0 yields the LIS algorithm). We used the uniform distribution as our proposal. We ran each sampler 50 times and computed the sample variance of the estimates.

**Results.** Figure 2 shows the sample variance of the estimators as a function of time. We see that the Rao-Blackwellised samplers typically have smaller variance than LIS . However, increasing the complexity bound typically does not improve the variance as a function of time (but the variance does improve as a function of number of samples). Our results indicate that the structure of the model plays a role in determining the most efficient complexity bound for sampling. In general, models with large decomposers, especially near the bottom of the schematic, will benefit from a larger complexity bound, because it is often more efficient to perform exact inference over a decomposer node.

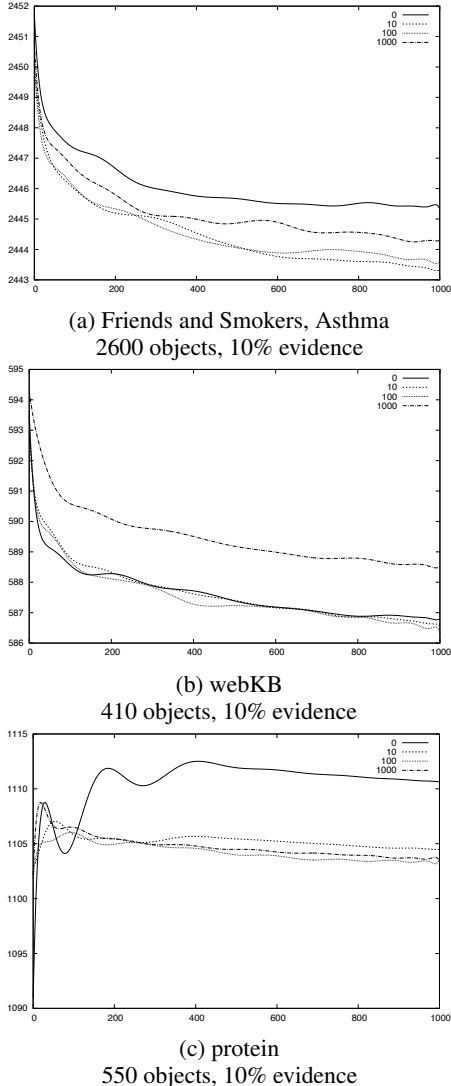

(a) Friends and Smokers, Asthma
2600 objects, 10% evidence

(b) webKB
410 objects, 10% evidence

(c) protein
550 objects, 10% evidence

Figure 2: Log variance as a function of time.

## 7 Conclusions and Future Work

In this work, we have presented an inference-aware representation of SRMs based on the And/Or framework. Using this framework, we have proposed an accurate and efficient method for bounding the cost of inference for the family of lifted conditioning based algorithms, such as Probabilistic Theorem Proving. Given a shattered SRM, we have shown how the method can be used to quickly identify tractable subproblems of the model. We have presented one immediate application of the scheme by developing a Rao-Blackwellised Lifted Importance Sampling Algorithm, which uses our bounding scheme as a variance reducer.

**Acknowledgments**

We gratefully acknowledge the support of the Defense Advanced Research Projects Agency (DARPA) Probabilistic Programming for Advanced Machine Learning Program under Air Force Research Laboratory (AFRL) prime contract no. FA8750-14-C-0005. Any opinions, findings, and conclusions or recommendations expressed in this material are those of the author(s) and do not necessarily reflect the view of DARPA, AFRL, or the US government.

## Footnotes

[1] Although, complexity bounds exist for related inference algorithms such as first-order decomposition trees [20], they are not as tight as the ones presented in this paper.

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
