[Supplementary Material]

# Bounding the Cost of Inference in Probabilistic Theorem Proving - Supplementary Material

## 1 Supplemental Material

Here we include additional discussion and experimental results not included in the paper. We give a proof of correctness of Algorithm CountPathLeaves (reprinted here as Algorithm 1 for convenience). We give the experimental results of log sample variance as a function of the number of samples (Figure 1). Additionally, in Section 3, we offer a more detailed discussion of the main complexity result in the paper.

## 2 Proof of Correctness for Algorithm CountPathLeaves

Here we prove that Algorithm CountPath-Leaves builds a function that correctly counts the leaves of a dependent counting path. The algorithm takes a dependent counting path as input (i.e. a totally ordered sequence of schematic nodes in which every lifted Or node counts over the same set of variables) and it returns a function, $f(x) : \mathbb{Z}^+ \to \mathbb{Z}^+$, that takes a domain size as input and returns the number of leaves corresponding to the dependent counting path given the input domain size. The algorithm works constructively by building a function for a path over $n$ decomposers in terms of the difference of the functions of paths over $n-1$ decomposers. Thus the leaf counting function of any dependent counting path is expressed in terms of the base case, a dependent counting path with zero decomposers. We consider this case first.

**2.1 Base Case-** First consider the base case: a dependent counting path over variable set $V$ (with domain $\Delta_V$) that contains $t$ total lifted Or nodes and no decomposers.

**Theorem 2.1** *Let $P$ be a dependent counting path containing $t$ lifted Or nodes and no decomposer labeled edges. Let $\Delta_V$ be the size of the domain of the variable set. Then the lifted Search space associated with $P$ contains $WC(\Delta_V, 2^t)$ leaves.*

---

**Algorithm 1** Function countPathLeaves

1: **Input:** a subsequence path $P$
2: **Output:** $f(x) : \mathbb{Z}^+ \to \mathbb{Z}^+$, where $x$ is a domain size and $f(x)$ is the number of search space leaves generated by $P$
3: //we represent the recursive polynomial `a(wc1 - wc2)` as a triple `(a,wc1,wc2)`, where `a ∈ ℤ`, and `wc1,wc2` are either weak compositions (base case) or triples of this type (recursive case)
4: **type** WCP = WC INT | WCD (INT,WCP,WCP)
5: `evalPoly` constructs the polynomial in WCP form
6: **function** MAKEPOLY((WC $n$), $(t,a,s)$)
7:     **return** WCD $(\frac{n}{2^{t-a}}, \text{WC } n, \text{WC } (n - 2^{t-a}))$
8: **end function**
9: **function** MAKEPOLY((WCD $(c, wc_1, wc_2)$), $(t,a,s)$)
10:     **return** WCD($a$, makePoly $wc_1$ $(t,a,s)$, makePoly $wc_2$ $(t - s, a - s, s)$)
11: **end function**
12: //applyDec divides out the Or nodes with counting variables that are decomposers
13: **function** APPLYDEC(d,(WC a))
14:     **return** WC $(a/(2^d))$
15: **end function**
16: **function** APPLYDEC(d,(WCD (a,b,c)))
17:     **return** WCD (a,applyDec d b,applyDec d c)
18: **end function**
19: //evalPoly creates a function that takes a domain and computes the differences of the constituent weak compositions
20: **function** EVALPOLY((WCD (a,b,c)),x)
21:     **return** a * (evalPoly b x - evalPoly c x)
22: **end function**
23: **function** EVALPOLY((WC a),x)
24:     **return** $\binom{x+a-1}{a-1}$
25: **end function**
26: t = totalOrNodes($P$)
27: dv = orNodesWithDecomposerCountingArgument($P$)
28: poly = WC $2^t$; orNodesAbove=0;orNodesBetween=0
29: **for** $N$ of $P$ **do**
30:     **if** $N = (A, \langle v, p, c \rangle)$ **then**
31:         poly = makePoly poly (t,orNodesAbove,orNodesBetween)
32:         orNodesBetween=0
33:     **else**
34:         orNodesAbove++;orNodesBetween++
35:     **end if**
36: **end for** **return** $2^{dv} *$ evalPoly (applyDec dv poly)

---

Proof: There are $2^t$ possible truth assignments to the $t$ nodes; thus the number of possible counting assignments is the number of ways to choose $2^t$ non-negative numbers that sum to $\Delta_V$. Algorithm 2 (evalNode(Or)) generates a leaf for each counting assignment. This quantity is the number of weak compositions of $\Delta_V$ into $2^t$ parts

(which we denote by $WC(\Delta_V, 2^t)$, as described in Equation (2)). Thus the dependent counting path generates $WC(\Delta_V, 2^t)$ leaves.

## 2.2 Constructive Case - Decomposer as Non-Counting argument-

Now consider the case in which the dependent counting path contains a single decomposer-labeled edge with $a$ lifted Or nodes above it and $t - a$ lifted Or nodes below it that do not count over the decomposer variable. Algorithm 1 (evalNode(And)) generates $WC(\Delta_V, 2^a)$ counting assignments above the decomposer label. At the decomposer label, each counting assignment (consisting of $2^a$ parts) decomposes into $2^a$ assignments (each consisting of a single part). Each non-zero decomposed assignment spawns a decomposed subtree that counts over the original assignment. Thus, each weak composition spawns a subtree for each of its non-zero parts. We must count the number of leaves generated by only the non-zero parts of each weak composition.

**Theorem 2.2** *Let $P$ be a dependent counting path containing a decomposer-labeled edge with $a$ lifted Or nodes above it and $t - a$ lifted Or nodes below it that do not count over the decomposer variable. Let $\Delta_V$ be the size of the domain of the variable set. Then the lifted Search space associated with $P$ contains $2^a(WC(\Delta_V, 2^t) - WC(\Delta_V, 2^t - 2^{t-a}))$ leaves.*

Proof: First, consider the number of leaves generated by decomposition of only the first of the $2^a$ elements of each weak composition. With no decomposer label, the number of leaves is $WC(\Delta_V, 2^t)$; we wish to subtract out those leaves that have the first term equal to $0$ after $a$ lifted Or nodes. The $0$ term will split into $2^{t-a}$ zero terms over the remaining $t - a$ nodes; thus, we can count the number of leaves with the first element of its counting assignment equal to $0$ after $a$ nodes as $WC(\Delta_V, 2^t - 2^{t-a})$. Therefore, we can represent the number of leaves with non-zero first element after $a$ lifted conditioning assignments as $WC(\Delta_V, 2^t) - WC(\Delta_V, 2^t - 2^{t-a})$. This relationship holds for each of the $2^a$ positions in the weak compositions above the decomposer; the number of leaves in the decomposed model equals $2^a(WC(\Delta_V, 2^t) - WC(\Delta_V, 2^t - 2^{t-a}))$.

(a) Friends and Smokers, Asthma
2600 objects, 10% evidence

(b) webKB
410 objects, 10% evidence

(c) protein
550 objects, 10% evidence

Figure 1: Log variance as a function of number of samples, for various Rao-Blackwellised estimators.

Thus the dependent counting path with a single decomposer label is represented in terms of a multiple of the difference of two dependent counting paths with no decomposers. Constructing the leaf counting function for paths with $n$ decomposers requires recursively applying this splitting rule to each weak composition in the expression for $n - 1$ decomposers. Lines $6 - 11$ of Algorithm 1 detail this procedure.

## 2.3 Paths with Decomposer Variables as Counting Arguments-

Dependent counting paths that contain lifted Or nodes that count over variables that have been decomposed on will generate fewer leaves, because the collection of $d$ lifted Or nodes with a decomposer counting variable counts over a domain of $1$ instead of $\Delta_V$;

| Model \c | 0 | 10 | 100 | 1000 |
|---|---|---|---|---|
| FSCA | 0.0266 | 0.0083 | 0.0061 | 0.0467 |
| WebKB | 0.0250 | 0.0371 | 0.0559 | 1.0218 |
| Protein | 0.0213 | 0.0361 | 0.1130 | 0.3523 |

Table 1: Time per sample for various complexity bounds on several models.

We can modify the leaf counting function detailed in 2.2 by dividing out the additional leafs.

**Theorem 2.3** *Let $P$ be a dependent counting path containing $t$ lifted Or nodes, a decomposer-labeled edge with $a$ lifted Or nodes above it and $t - a$ lifted Or nodes below. let $d$ be the number of lifted Or nodes that count over $V$ below the decomposer label. Let $\Delta_V$ be the size of the domain of the variable set. Then the lifted Search space associated with $P$ contains $(2^{d+a})(WC(\Delta_V, 2^{t-d}) - WC(\Delta_V, 2^{t-d} - 2^{t-a-d}))$ leaves.*

Proof: Each lifted Or node with a decomposer variable as its counting variable counts over a domain of 1 instead of $\Delta_V$, and hence will generate $2^d$ leaves instead of $WC(\Delta_V, 2^d)$ leaves. Thus the number of leaves generated by these Or nodes is independent of the domain $\Delta_V$. Thus we can remove those nodes from the formula given in Theorem 2.2 by dividing the number of parts of each weak composition by $2^d$, and then multiplying the resulting function by $2^d$. The procedure is detailed in lines $13 - 18$ of Algorithm 1.

## 3    Additional Complexity Results

Here we offer a more detailed discussion of Theorem 3.1, the primary complexity result of the paper:

**Theorem 3.1** *Given a lifted And/Or Schematic $S$ with associated Tree Decomposition $D_S = (\mathcal{C}, T)$, the overall time and space complexity of inference in $S$ is $O(max_{C_i \in \mathcal{C}} SS_C(C_i))$.*

This result follows from the fact that $SS_C$ correctly counts the number of leaves generated by any cluster. In terms of space complexity, it suffices to cache only the leaf nodes of each cluster, so the space complexity result given in Theorem 3.1 is a strict upper bound; with a little more work, we can state the space complexity exactly.

**Corollary 3.2** *Given a lifted And/Or Schematic $S$ with associated Tree Decomposition $D_S = (\mathcal{C}, T)$, the overall space complexity of inference in $S$ is $\sum_{C_i \in \mathcal{C}} SS_C(C_i)$.*

Similarly, we can compute a more precise expression for time complexity by computing the exact number of copies of each node $N \in Nodes(S)$ produced by Algorithm $evalNode$. To compute the number of leaves produced for a node $N$, we must (1) find the set $\mathcal{C}_N$ of all clusters $C_i$ s.t. $N \in C_i$, (2) for each $C_i = \{N_1, \ldots, N_k\}$ extract the path $P_i^N = \{N_1, \ldots, N\}$, and (3) compute $SS_N(N) = max_{C_i \in \mathcal{C}_N} SS_C(P_i^N)$. Given this information, the exact number of leaves in the search space produced by Algorithm $evalNode$ on $S$ simply a sum of the copies its produces of each node.

**Corollary 3.3** *Given a lifted And/Or Schematic $S$ with associated Tree Decomposition $D_S = (\mathcal{C}, T)$, the overall time complexity of inference in $S$ is $\sum_{N_i \in Nodes(S)} SS_N(N_i)$.*