[Reviews · NeurIPS 2015]

Submitted by Assigned_Reviewer_1

The paper introduces an approach to estimate the partition function when doing lifted inference in statistical relational models. The paper is well written and clear and proposes a lifted version of the And/Or search space framework that, to the best of my knowledge, is novel.

In general, the ideas presented in the paper are sound.

Nevertheless, the evaluation is a bit short and it fails to compare their approach to other approximations of the partition function. The authors do not include an analysis of the bounds of the approximation and a study of how it scales in general.

It is possible that their lifted And/Or framework can provide good reductions in the complexity of inference but their evaluation and results are too limited to tell. Nevertheless, this paper does introduce a novel algorithm that may prove useful for the NIPS community interested in lifted inference.

Summary: In general, the ideas presented in the paper are sound. Nevertheless, the evaluation is a bit short and it fails to compare their approach to other approximations of the partition function.

Submitted by Assigned_Reviewer_2

The paper introduces lifted and/or schematics, which are an abstract representation of the PTP search space. Analyzing the schematic lets us quantify the size of the search space. This is of independent theoretical interest (although the theory is not quite there yet to provide deep insights), and is practically useful in the context of Rao-Blackwellized importance sampling.

I believe this type of work is important; we still lack understanding of lifted inference algorithms. The proposed schematics are not really simple enough to provide a whole lot of insight, but it's a move in the right direction. The presentation is okay, although I cannot verify many of the technical points with the time available to me.

Given that this is a theoretical paper, I wonder how powerful the framework is in terms of liftability. Can these schematics capture all known classes of models that are domain-liftable?

The paper misses the opportunity to relate to other similar techniques.

How are dtrees (used in recursive conditioning and knowledge compilation) related to and/or schematics?

How are first-order NNF circuits related to lifted and/or schematics? They look very similar, with counting nodes having a single child, which induces a summation when evaluated, and so on. There is some work on the complexity of evaluating partition functions from first-order NNF circuits, where the result is that it is exponential in the number of nested counting operations (existential quantifiers). This result seems related to the analysis in Section 4.

How does this relate to the first-order decomposition trees of Taghipour et al.? The footnote on page 3 says that these are not as tight: I would expect more details here (and perhaps an example of where the complexity differs).

I like the importance sampling algorithm that uses the theory in a meaningful way. It's very convincing. I know these experiments are preliminary, but I would still like more details on the setup. For example, I do not believe that 10% evidence on webkb leaves you with any symmetries to exploit. I suspect that the evidence predicates were chosen to maximize the remaining symmetries (only evidence on page class, not on links?). It would be good to clarify.

Typos: The theta_i on line 105 should be a t_i. With exchangeable variables on line 171, you mean exchangeable constants (the random variables after shattering are not fully exchangeable in general). Line 212: of depends.

Summary: This is good work on a very hard problem. The theoretical analysis is very difficult to follow but makes sense at a high level, when you know the lifted inference literature. The sampling application is cute and makes the theory immediately useful.

Submitted by Assigned_Reviewer_3

This is a paper on the line of combining lifted and approximate

inference. In this case the two contributions are: estimates of the computional effort , and using an a Rao-Blackwellised lifted importance sampling. The text is well written, but assumes good understanding of previous work, namely ref [9] comes up all the time. I'd also like some more on RaoBlackwell. Experimentsl evslustion is weak.

46-47 I understand the concept of inference unaware, but the idea that being aware of inference means having an idea of how much computation ahead is strange for me, because usually we just can't.

54 - a traditional and-or tree is not a "compact"representation. What do you by "pseudotree"? Folded tree?

57 - ok, I got

the idea, but why do you call it schematic?

81-90 there is an and/or tree, which is a spanning tree for a graph that is the original graphical model, there is a mysterious pseudo-tree, there is anther final graph, I think I understnd but please try to define precisely what is what.

119 ->new MLNs ? what do you define as a MLN?

148-160 -> these are complex rules, should have a minimal description. Also explain decomposer,just citing [9] is not enough?

176 - optimal? Is there an optimal?

212 "The overall complexity of depends:"

missing word

346 - which in turn decreases the 347 accuracy because fewer samples are gener- 348 ated.

-> because time budget is fixed?

Sec 6 You previously said: "We demonstrate experimentally that it vastly improves the accuracy of estimation on several real-world datasets." Is this a valid conclusion from your results?

Summary: This is a paper on the line of combining lifted and approximate

inference. In this case the two contributions are: estimates of the computional effort, and using an a Rao-Blackwellised lifted importance sampling. The paper focus on the first, although I think the second quite interesting too. In general it looks like solid, robust work, but I don't see how the experimental results suppport the claims.

Author Feedback
Author rebuttal: Thank you for the reviews; they help us improve the paper greatly.

Review1 : With regard to readability concerns, we acknowledge that the paper is dense. Should the paper be accepted, we will reformat the camera-ready version in order to make it more readable. We will relocate the inset algorithms and figures to the top. We can create space by moving subsubsection titles (for example, "4.3.1") to inline titles and simplifying the pseudo-code.

The pseudocode can be unintuitive. We struggled to make the algorithms easy to understand without sacrificing correctness. We will revisit these sections and simplify again; where we cannot simplify we will give more relevant names and add clarifying comments in the text rather than the pseudo-code.

While the experimental results are preliminary or proof-of-concept, we think that they accurately reflect the fact that Rao-Blackwellisation must be used with caution. While it guarantees a decrease in variance over n samples, it also increases the complexity of generating each sample. So fewer samples can be taken within a set time limit. We reported variance as a function of time in the main paper in order to show that the algorithm did indeed reduce variance as a function of time (which is the most useful in practice) on models for some choices of sample complexity bound. We also included results on variance as a function of samples in the supplemental material in order to show the guarantee on variance reduction. We generated random evidence across all atoms (and so unlike other exact work on lifted inference, there is no special selection of evidence atoms). We will tone down the language used to describe the results. Notice that we cannot compute the exact value of the partition function and therefore including other partition function approximations (e.g., BP-based) is not relevant here because these approximations are deterministic and there is no notion of variance.

We felt that our main contribution was the complexity result because of its utility in a range of approximation schemes for SRMs. For example, we are currently investigating its use in (1) an ordering algorithm for SRMs and (2) a Generalized BP algorithm that relies on the result in order to ensure tractable clusters. We could have easily written a paper just on Rao-Blackwellisation. However, we want to keep the presentation general for future applications.

Review 3
By 'inference aware,' we mean a representation that includes all the information needed to compute complexity at any given step. For example, given a PGM, computing the best runtime of the variable elimination algorithm is NP-hard. However, given an ordering (chosen heuristically) we can compute the complexity in P-time. Similarly, the PTP algorithm makes heuristic choices during execution that affect its runtime. The point of our lifted And/Or schematic is to represent this information. We refer to it as a schematic because it compactly encodes all the information needed by PTP, thus guiding the execution of the algorithm. In this way it is analogous to the pseudotree concept in the And/Or literature.

Review 4
Our goal was to define a model that permits analysis of both kinds of symmetry breakers in SRMs: (1) those caused by logical structure, and (2) those caused by evidence. In order to find tight complexity bounds, we chose to focus on a specific algorithm (PTP with context minimal caching), and we chose to adopt the And/Or terminology because of our familiarity with that line of research. Thus the framework is able to express all known domain liftable classes of models (it has the same complexity as PTP), and the size of the schematic is independent of the domain size exactly when PTP can lift over every domain. The framework is similar to FO-dtrees and the Big O complexity results can be extended to them as well. We will include two sentences on this in a camera ready version of the paper, should it be accepted. However, our results are novel as explained below.

We were unable to find a closed form expression for complexity as a function of some sufficient statistic of the model; instead we devised an algorithm that computes a polynomial which takes domain sizes as input and returns the number of times a given node in the model will be replicated in the search space. The degree of the polynomial is exponential in the structural properties of the schematic (the size of the dependent counting group of the logical variables). It is worth noting that the polynomial computes this number exactly; hence it can be used to get the exact number of nodes in the search space (Corollary 3.3 in the supplemental material), which distinguishes the work from that of Taghipour et al (see Theorem 2 in Taghipour). The difference can be enormous, making our result more useful in practical applications. We believe that in future, our results can be used as an analytical tool to glean insights of theoretical interest.